# MANAR: Memory-augmented Attention with Navigational Abstract conceptual Representation

## Abstract

Transformers - and their multi-head attention (MHA) core - power today's leading models across a broad application spectrum. Yet MHA contextualizes each token through explicit, pair-wise interactions with every other token, yielding quadratic time/space cost and an unbounded, linearly-growing context. This *direct all-to-all* modeling is both the source of attention's expressiveness and a barrier to scaling. We address this bottleneck by augmenting attention with a trainable external memory that stores both conceptual and relational general representations learned during training. For every input, lightweight scalable retrieval produces a fixed-size set of memory retrieved concepts whose values are fused into a compact Abstract Conceptual Representation (ACR). Tokens then attend jointly to (i) the global, concept-level ACR and (ii) a short local context, completely sidestepping all-to-all token interactions. The result is a non-convex pathway that provides the model with an *out-of-the-box thinking* contextualization - i.e., beyond the convex hull spanned by the input values - while reducing complexity to linear time and memory. Integrated as a drop-in replacement for MHA, our layer (MANAR) preserves accuracy on ImageNet-1K (82.3% top-1 with a DeIT-B backbone) and LibriSpeech (2.9/6.8% WER on test-clean/other) yet cuts inference latency by up to 14.8x and peak GPU memory by up to 9.3x as sequence length grows to 4K. A simple weight-copy knowledge-transfer procedure trims training cost by ≈99% versus training from scratch. Finally, Convex Hull Membership (CHM) tests show that >50% of MANAR's outputs lie outside the convex span of the input values, quantitatively confirming its out-of-the-box contextualization.

## 1 Introduction

Since its introduction, the transformer architecture (Vaswani et al., 2017) has achieved remarkable success across a wide spectrum of domains, including natural language processing (Vaswani et al., 2017; Karpukhin et al., 2020; Touvron et al., 2023; Liu et al., 2024a; Warner et al., 2024), computer vision (Dosovitskiy et al., 2020; Arnab et al., 2021; Shehzadi et al., 2023), speech recognition (Schneider et al., 2019; Baevski et al., 2022; Liu et al., 2023), bioinformatics (Brandes et al., 2022; Acera Mateos et al., 2021; Jahshan & Yavits, 2024), and many other. At the heart of this success lies the attention mechanism, which enables every token to attend to all other tokens in a sequence, yielding highly expressive, sequence-wide contextualizations and facilitating efficient parallel training. However, the same all-to-all contextualization that powers the expressive capability of MHA introduces significant scalability bottlenecks. The quadratic time and memory complexity with respect to sequence length—together with the need to store linearly growing, unbounded context in autoregressive generation—limits both the efficiency and the reach of contemporary attention-based models. This challenge has become increasingly pressing as workloads shift toward longer contexts and larger-scale models, motivating numerous strategies to address these limitations.

The most widely adopted approaches, such as quantization and knowledge distillation, have succeeded in compressing the memory and compute footprint of transformer models. Quantization (Ashkboos et al., 2024; Liu et al., 2024c;d; Xiao et al., 2023) enables more efficient computations and reduced memory footprint by lowering precision, sometimes as low as 4-bit per element (Liu et al., 2024c). Distillation (Bing et al., 2025; Han et al., 2024; Mukherjee et al., 2021) transfers knowledge from large models to smaller ones. Despite their practical utility, these methods fall short of solving the core issue: the direct all-to-all token contextualization is preserved, so context size remains unbounded and computational complexity quadratic. Orthogonally, efforts to directly modify the attention mechanism have led to sub-quadratic architectures. Linformer (Wang et al., 2020) experimentally show that the attention matrix present a low-rank behavior, hence it projects the keys and values into lower-dimensional spaces, allowing attention computation to operate in linear time and space. Performer (Choromanski et al., 2020) employs kernel-based random feature projections to approximate softmax attention, similarly achieving linear scaling. "Transformers are RNNs" (Katharopoulos et al., 2020) conceptualizes the transformer as a recurrent model, allowing for incremental, stepwise computation, and efficient reuse of memory across timesteps.

To solve the unbounded context problem, many works (Xiao et al., 2024; Fountas et al., 2025; Sun et al., 2024; Lee et al., 2024; Liu et al., 2024b; Mohtashami & Jaggi, 2023; Wu et al., 2022) offload the KV cache into lower memory tiers (e.g. CPU DRAM), and sparsely select KV pairs to attend to. These techniques proved very useful, partly because of the observation that when contextualizing one token, many tokens do not actually get involved in the process due to very low attention score. To take advantage of this behavior, some of the aforementioned works augment the multi-head attention with a memory unit, saving contextual blocks (i.e., KV blocks) to later retrieve them upon contextualization. For example, InfLLM (Xiao et al., 2024) contextualizes each token through its local context window as well as highly related selected KV blocks. Upon token contextualization, the memory unit is queried by the token to retrieve highly relevant KV blocks in a k-nearest-neighbor fashion. Both the retrieved blocks and the token local context window participate in the process of token contextualization.

Many works seek to completely replace the standard attention mechanism with alternative architectures designed for enhanced scalability. For example, Titans (Behrouz et al., 2024) and ATLAS (Behrouz et al., 2025) augment standard attention (used as short-term memory) with explicit long-term neural memory modules, which are updated using test time training and optional persistent memory to retrieve and fuse past information alongside current context attention for scalable long-range modeling. State space models (Gu et al., 2021) formulate sequence modeling as linear dynamical systems, allowing fast, parallel computation, and effective modeling of long-range context. Mamba (Gu & Dao, 2023) extends this paradigm by introducing selective long-range sequence modeling through dynamic state selection and gating, enabling efficient and expressive representations for extremely long inputs (Dao & Gu, 2024). Other notable advances, such as RetNet (Sun et al., 2023), propose recurrent architectures with retention mechanisms that further boost performance, demonstrating state-of-the-art results on long-context benchmarks. These approaches fundamentally rethink the model design for sequential data and show particular promise in ultra-long sequence regimes.

We introduce *MANAR*, a brain-inspired, memory-centric attention architecture functioning as a contextualization layer to be plugged in a commodity transformer encoder models. The architecture draws inspiration from cognitive processes in the human brain, wherein perception and comprehension depend not only on inputs but also on internalized, generalized concepts built from prior experience. When presented with an external input, the brain builds a mental image depending on previous memorized concepts associated with the observed input, and the way it perceives their connection. This mental image, along with recently observed inputs, which are remembered, guides and navigates contextualization - the process in which meaning are given to each observed input occurrence. We also demonstrate that when comprehension is based on memorized abstract concepts, a behavior we term *out-of-the box thinking* can be distinguished.

Specifically, when presented with a sequence of tokens, MANAR (i) retrieves memory concepts that help constructing a full-context-wide, constant-sized abstract conceptual representation functioning as a mental image, aiming at representing the 'global abstract themes' present in the sequence, and (ii) contextualize each input token navigated by this mental image as well as the local context window of each token, avoiding direct all-to-all contextualization. MANAR can be integrated as a drop-in replacement for standard MHA layers, making it practically deployable in existing transformer encoder stacks. Empirical evaluation in large-scale image classification and automatic speech recognition demonstrates that MANAR matches or exceeds the state-of-the-art: for example, it achieves 82.3% top-1 accuracy on ImageNet-1K (Deng et al., 2009) and 2.9/6.8% WER on LibriSpeech's test-clean/other sets (Panayotov et al., 2015) with significant inference speedups (up to 14.8x) and peak GPU memory reductions (up to 9.3x) as sequence length increases. A straightforward weight-copy knowledge transfer procedure enables rapid adaptation from pretrained transformers, reducing training time by 99% compared to training from scratch. Convex Hull Membership (CHM) tests show that over 50% of MANAR's outputs lie outside the convex hull of input token values—a rigorous demonstration of its out-of-the-box contextualization capability. We make our code available as a part of supplementary material.

## 2 PRELIMINARIES AND BACKGROUND

First, we define the notion of a concept and the concept contextualization process as follows:

**Definition 2.1** (Concept). A concept $x$ is defined as a tuple $x = (q, k, v)$. The concept query, $q$, the concept key, $k$, and the concept value, $v$.

**Definition 2.2** (Contextualization). Given a concept $x = (q, k, v)$, and a set of concepts $C = \{(c_1^q, c_1^k, c_1^v), \ldots, (c_n^q, c_n^k, c_n^v)\}$ we define the contextualization process of $X$ by C as:

$$y = \sum_{(c^q, c^k, c^v) \in \tilde{C}} S(q, c^k) c^v$$

Figure 1: A: Black dots represent input token values. The gray space represents these points' convex hull, which spans the image of the output. B: Black dots represent the input token values while the red dot represents a retrieved memory value. The area in red and black represents the image of the output when contextualization is coupled with a retrieved memory value.

$$\text{where } \tilde{C} = C \cup \{x\}; \qquad S(\cdot, \cdot) \geq 0; \qquad \sum_{(c^q, c^k, c^v) \in \tilde{C}} S(q, c^k) = 1$$

For each concept, the query representation is used to decide how this concept is contextualized when it is linked to another set of concepts. The key representation is used to decide how this concept influences the contextualization process of other concepts. The value representation is a vector holding the meaning of the concept. The contextualization process influences the meaning of the contextualized concept, $x$, and shifts it towards a point located in the meaning space spanned by the convex hull of $\{c^v : (c^q, c^k, c^v) \in \tilde{C}\}$, a process depicted in Fig[1](a).

The multi-head attention (MHA) block is a central component of the Transformer architecture, enabling each layer (encoder or decoder) to map an input sequence to a same-length sequence of contextualized outputs in which every position can integrate information from all input tokens. MHA conceptualizes every input token by projecting them into query, key, and value representations, and contextualize them by all other conceptualized input tokens. Formally, let the input be $X = \{x_1, ..., x_n\} \in \mathbb{R}^{n \times d}$ and let $W_q, W_k, W_v \in \mathbb{R}^{d \times d}$ be learnable projections; define $q_i = x_i W_q, k_i = x_i W_k, v_i = x_i W_v$, and compute

$$y_i = \sum_{j=1}^{n} S_{i,j} v_j = \sum_{j=1}^{n} \frac{e^{q_i k_j^T}}{\sum_{l=1}^{n} e^{q_i k_l^T}} v_j \qquad (1)$$

where $S_{i,j}$ denotes the attention weight induced by the softmax over query–key similarities. This contextualization is powerful partly because it avoids explicit spatial or temporal biases, allowing dependencies across arbitrary positions; information flows to the $i$-th output proportionally to its similarity to the $i$-th query (i.e., the magnitude of $q_i k_j^T$). Nevertheless, such contextualization makes the time and memory cost of full attention scale quadratically with sequence length, which limits scalability to long inputs. Moreover, to make this contextualization process possible, the entire $kv$ state must be persisted in autoregressive tasks.

In this work we aim at developing a different contextualization process. When an external sequence of inputs is presented, internal memorized concepts highly associated with the input are retrieved (an idea inspired by human brain). Links and associations are then made between input tokens and retrieved memorized concepts. These links take the shape of some constructed global conceptual representation of the perceived input. This constructed perceived representation then guides and navigates the contextualization process of the presented external inputs. We can examine the human reading comprehension process as an example of this behavior. When presented with an input text, the text read stimulates the brain to retrieve some memorized concepts associated with it. Throughout the reading process, brain constructs a mental image - abstract conceptual representation - representing the perception of what is being read and its connections to memory. Then, every word in the text is contextualized and understood in light of this constructed perceived abstract representation (Kewenig et al., 2024; Keller et al., 2024).

## 3 MANAR

MANAR contextualizes input tokens by neighboring tokens as well as by retrieved memorized concepts not present in the input sequence. To make it possible, MANAR integrates a memory unit that retains memorized concepts. When an input sequence of tokens $X$ is presented, $m$ search patterns are generated and applied to retrieve $m$ memorized concepts from the memory unit in a fast and scalable manner. After retrieval, these internal memory concepts are linked and associated to the presented input forming the Abstract Conceptual Representation ($ACR$) which functions as a mental image. Then, input tokens are contextualized by the $ACR$ as well as the local context window of the token to formulate the output of the layer.

### 3.1 NOTATIONS

We start by discussing some notations that are used consistently throughout the paper. Let $X \in \mathbb{R}^{n \times D}$ represent the input sequence of tokens, where $n$ is the sequence length and $D$ is the dimension of each observed input token. Let $\mathcal{M}_i$ be the $i$-th retrieved memory concept which is consistently represented as a qkv-tuple $\mathcal{M}_i = (c_i^q, c_i^k, c_i^v)$. Use $m$ to refer to the number of retrieved memory concepts, and $M$ to refer to the total memory size (i.e., the total number of memory cells in the memory unit). The $ACR$ size is $m \times d$ (where $d$ is the per-head dimension), a row for each retrieved memory concept. We refer to the $i$-th row of the $ACR$ by $ACR_i$ or $r_i$ interchangeably. We refer to $d$ as the intra-layer dimension (i.e., per-head dimension), and we assume $D = hd$ where $h$ is the number of heads. Lastly, the $i$-th output (i.e, the contextualized $i$-th token) is referred to as $y_i$. In this work we follow a row-major representation.

### 3.2 ACR CONSTRUCTION AND TOKEN CONTEXTUALIZATION

Throughout Sec. 3.2 we assume the existence of $m$ retrieved memory concepts $\left\{\mathcal{M}_i = (c_i^q, c_i^k, c_i^v)\right\}_{i=1}^m$ decoupling the process of memory retrieval from the rest of the MANAR architecture. Memory retrieval is discussed separately in Sec. 3.3. Calculations are made considering a single-head architecture. Multi-head architecture generalization is made in section. 3.4.

MANAR defines four learnable projection matrices $W_q, W_k^{\mathcal{M}}, W_k, W_v \in \mathbb{R}^{D \times d}$ corresponding to the token's query, ACR key, contextualization key, and value, respectively. $W_k^r \in \mathbb{R}^{d \times d}$ represents the projection responsible for converting $ACR$'s into "token contextualization" key-space. Moreover, we define the Region of Interest of an index $i$ with a neighborhood $l$ as:

$$ROI^l(i) = \{j : max(0, i - l + 1) < j \leq min(n, i + l)\}$$

where $ROI^l(i)[j]$ represents the $j$-th smallest element in the set. The $ROI$ is used to represent the local context window in the process of token contextualization. We refer to local context window length as the maximal size of the $ROI^l$ which is $2l$. Equipped with these learnable parameters and the $ROI$, the logic of the MANAR layer operating in two stages, the ACR construction and the token contextualization, is defined as follows:

- Conceptualization:

$$k_i^{\mathcal{M}} = x_i W_k^{\mathcal{M}}; \qquad q_i = x_i W_q; \qquad k_i = x_i W_k; \qquad v_i = x_i W_v \qquad (2)$$

- Stage 1 (ACR Construction):

$$r_i = S_{i,0} c_i^v + \sum_{j=1}^n S_{i,j} v_j \qquad (3)$$

$$\text{where } (S_{i,0}, S_{i,1}, S_{i,2}, \ldots, S_{i,n}) = softmax\left(\frac{c_i^q \cdot (c_i^k)^T}{\sqrt{d}}, \frac{c_i^q \cdot (k_1^{\mathcal{M}})^T}{\sqrt{d}}, \ldots, \frac{c_i^q \cdot (k_n^{\mathcal{M}})^T}{\sqrt{d}}\right) \qquad (4)$$

The process of ACR construction can be seen as contextualizing each retrieved memory concept by the conceptualized input tokens. Concretely, the $i$-th ACR vector, $r_i$, represents the meaning of the memorized concept shifted according to how strongly that memory concept associates with each observed token. The strength of an association is measured by the inner product $c_i^q \cdot (k_j^m)^T$. Intuitively, this mirrors human cognition: our mental image of a situation blends internal memorized concepts with incoming evidence, weighted by the perceived relevance of each piece of evidence to those concepts.

- Stage 2 (Token Contextualization):

$$y_i = \underbrace{\sum_{j=1}^m \hat{S}_{i,j} r_j}_{\text{global attention}} + \underbrace{\sum_{j=1}^L \tilde{S}_{i,j} v_{ROI^l(i)[j]}}_{\text{local attention}} \qquad (5)$$

$$\text{where } (\hat{S}_1, \ldots, \hat{S}_m, \tilde{S}_1, \ldots, \tilde{S}_L) = \qquad (6)$$

$$softmax\left(\frac{q_i(r_1 W_k^r)^T}{\sqrt{d}}, \ldots, \frac{q_i(r_m W_k^r)^T}{\sqrt{d}}, \frac{q_i k_{ROI^l(i)[1]}^T}{\sqrt{d}}, \ldots, \frac{q_i k_{ROI^l(i)[L]}^T}{\sqrt{d}}\right)$$

$$\text{and } L = |ROI^l(i)|$$

Navigated by the constructed $ACR$ serving as a mental image, MANAR contextualizes each input token in light of this mental image, as well as all the input information that could be perceived at once (i.e., the local context window), represented by the $ROI$ of the token. Hence, the meaning each contextualized token holds is influenced both by association with the mental image and with neighboring tokens.

Since MANAR's mental image is constructed around memorized concepts and the meaning they hold, and also since the process of token contextualization is navigated by this mental image, MANAR-contextualized tokens can obtain meanings not bounded to the meaning space spanned by the convex hull of inputs $\{xW_v : x = X_i, 1 \le i \le n\}$. We refer to this contextualization behavior as *out-of-the-box thinking*. The out-of-the-box thinking is expressed by the following derivation of $r_i$:

$$r_i = S_{i,0}c_i^v + (1 - S_{i,0}) \sum_{j=1}^{n} \frac{S_{i,j}}{1 - S_{i,0}} v_j \tag{7}$$

$$= S_{i,0}c_i^v + (1 - S_{i,0}) \sum_{j=1}^{n} \frac{\frac{e^{c_i^q (k_j^m)^T}}{e^{c_i^q (c_i^k)^T} + \sum_{l=1}^{n} e^{c_i^q (k_l^m)^T}}}{\frac{\sum_{l=1}^{n} e^{c_i^q (k_l^m)^T}}{e^{c_i^q (c_i^k)^T} + \sum_{l=1}^{n} e^{c_i^q (k_l^m)^T}}} v_j \tag{8}$$

$$= S_{i,0} \underbrace{c_i^v}_{B} + (1 - S_{i,0}) \underbrace{\sum_{j=1}^{n} \frac{e^{c_i^q (k_j^m)^T}}{\sum_{l=1}^{n} e^{c_i^q (k_l^m)^T}} v_j}_{A} \tag{9}$$

Eq. 9 demonstrates that every ACR is a weighted sum of two expressions: (A): Expression having the same form of the output expression produced by the MHA layer as appearing in Eq. 1; (B): Correction expression, potentially shifting the ACR out of the input tokens convex hull, as illustrated in Fig. 1(B).

## 3.3 THE MEMORY UNIT

perates by two stages, the ACR Construction, and the Response Construction. In this section, we discuss the memory unit, and the memory retrieval process of $m$ concepts, $\mathcal{M}_i = (c_i^q, c_i^k, c_i^v)$, completing the full picture of MANAR.

The memory unit contains $M$ memory cells. Each memory cell retains a concept, $\boldsymbol{\mu}_i = (\mu_i^q, \mu_i^k, \mu_i^v)$, where $0 < i \le M$. The memory retrieval process involves the creation of $m$ different search patterns as a function of input tokens. For each search pattern, top-$k$ memory cells are chosen on the basis of their similarity to the search pattern. The memory concept is calculated as a weighted sum of the contents of these matching top-$k$ memory cells. The logic of producing the search pattern is first formalized, then we detail how each search pattern drives retrieval.

To perform $m$ memory lookups, the model first constructs $m$ search patterns from the input sequence $X \in \mathbb{R}^{n \times d}$. Concretely, the model introduces $m$ learnable "mixer" vectors $mixer_i \in \mathbb{R}^d$ that aggregate information from tokens via a cross-attention operation where the queries are the mixer vectors and the keys/values come from the tokens. Let $W_k^{SP}, W_v^{SP} \in \mathbb{R}^{d \times d}$ be learnable projections; the $i$-th search pattern is then:

$$\sigma_i = softmax \left( \frac{mixer_i \cdot \left(XW_k^{SP}\right)^T}{\sqrt{d}} \right) \cdot XW_v^{SP} \tag{10}$$

Given a search pattern $\sigma_i$, the memory unit keys, a table of keys, one per each memory cell, $\xi \in \mathbb{R}^{M \times d}$, and the memory cells $\boldsymbol{\mu} \in \mathbb{R}^{M \times 3d}$, retrieval computes a soft combination of cells weighted by their similarity to $\sigma_i$. The retrieval step is:

$$I = SelectTopkIndices(\sigma_i \cdot \xi^T); \qquad s = softmax(\sigma_i \cdot (\xi_I)^T); \qquad \mathcal{M}_i = s \cdot \boldsymbol{\mu}_I \tag{11}$$

where $I$ is a set of indices, $s \in \mathbb{R}^k, \xi_I \in \mathbb{R}^{k \times d}, \boldsymbol{\mu}_I \in \mathbb{R}^{k \times (3d)}$, and the output $\mathcal{M}_i \in \mathbb{R}^{3d}$.

Scaling the memory size, $M$, makes naive nearest-neighbor scoring over all keys $\mathcal{O}(M)$ per search pattern prohibitive; fast approximate similarity search techniques could be used here (Johnson et al., 2019), but incorporating them is challenging when keys are continually trained and re-indexed. MANAR uses trainable product keys (Lample et al., 2019), which factor the key space into two tables $\xi^{(1)}, \xi^{(2)} \in \mathbb{R}^{\sqrt{M} \times \frac{d}{2}}$ whose (implicit) Cartesian product spans M composite keys without materializing them. For lookup, split the search pattern as $\sigma_i = \left[\sigma_i^{(1)}; \sigma_i^{(2)}\right]$ with $\sigma_i^{(1)}, \sigma_i^{(2)} \in$

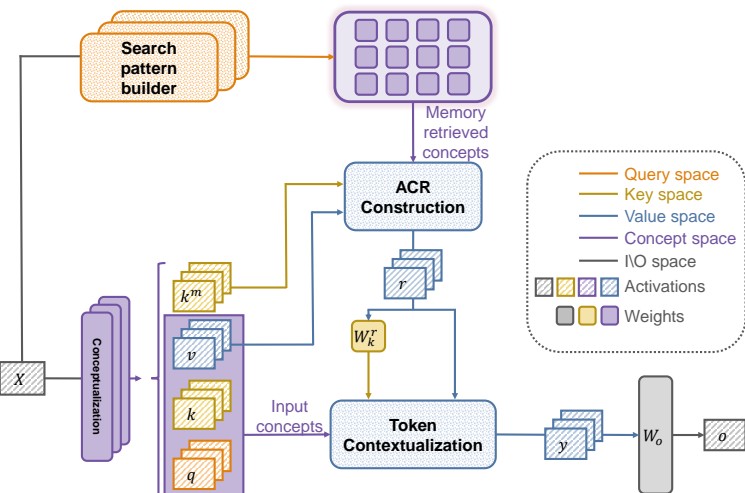

Figure 2: High level archWitecture of MANAR

$\mathbb{R}^{\frac{d}{2}}$, retrieve top-k indices and scores $(I_1, s_1)$ from $\xi^{(1)}$ and $(I_2, s_2)$ from $\xi^{(2)}$, then combine candidates by maximizing summed scores over pairs:

$$argmax_{j_1 \in I_1, j_2 \in I_2} s_1[j_1] + s_2[j_2]$$

which yields an efficient approximation to top-$k$ over the full $M$ composite keys while searching only the $\sqrt{M}$-sized half-key tables.

### 3.4 MULTI-HEAD ARCHITECTURE

To form a $h$-head MANAR layer we replicate the entire single-head conceptualization pipeline, including the search-pattern builder, one per head. Each head therefore learns its own token projections and search patterns, yet all heads reuse the same token-contextualization key projection $W_k^r$ and access one shared external memory. After each head completes retrieval, ACR construction and token contextualization, their outputs are concatenated and mapped back to $D$ dimensions through a single output projection, preserving the standard transformer interface. Fig. 2 depicts the entire multi-headed architecture. Moreover, a fully comutational analysis of the layer is provided in A.1.

## 4 EVALUATION

To evaluate MANAR accuracy, performance and memory usage, we apply it as a drop-in replacement to MHA in several transformer-based AI models. When comparing vanilla transformer encoder architecture to MANAR-enabled one, we leave all other layers unmodified. In this section, we refer to any MANAR-enabled transformer encoder architecture, having a memory of size $M$ and $ACR$ of size $m$ that aggregates top-$k$ memory cells to assemble a memory concept as "MANAR-$M.m.k$".

### 4.1 IMAGE CLASSIFICATION

We benchmark MANAR on the ImageNet-1K dataset (Deng et al., 2009), which contains 1.28M training images and 50K validation images from 1000 categories. We use DeiT-B and DeiT-S (Touvron et al., 2021) as our baselines, hence, we refer to MANAR-$M.m.k$-B(-S) as a DeiT-B(-S) transformer encoder where all MHA layers were replaced by MANAR layers. We trained a MANAR-256.32.8-B(-S) model with 12 encoder layers, each containing 12(6) heads. The dimensionality of each head was set to $d = 64$. For each layer we held a local context window attention attending to the 96 neighboring tokens. The model is trained on the training set and the top-1 accuracy on the validation set is reported. For fair comparisons, we trained the model from scratch with training settings used in DeIT. Specifically, we apply random cropping, random horizontal flipping, label smoothing regularization, mixup, and random erasing as data augmentations. The training took place on images of size $224^2$. We employ AdamW (Loshchilov et al., 2017) with a momentum of 0.9, a total batch size of 1024, and a weight decay of $5 \cdot 10^{-2}$ to optimize the model. We train the MANAR based DeiT architecture for 450(300) epochs using the cosine scheduling with a learning rate initiated as

$4 \cdot 10^{-4}$ and EMA. During testing we apply a center crop on the validation set to crop out $224^2$ images. Experiments are performed on a single H100 GPU. Top-1 accuracy on validation set results are reported in Tab. 1. A comparison of our architecture was made against models that have similar parameter count. Our architecture was compared to transformer based Deit-B(-S), as well as Vision Mamba, Vim-B(-S) (Zhu et al., 2024) which is a linear complexity architecture, and vanilla Vision Transformer (Dosovitskiy et al., 2020). As can be seen in the table, MANAR slightly outperforms models of approximately the same size, demonstrating its ability to achieve state-of-the-art accuracy performance.

Table 1: Comparison of different backbone architectures on ImageNet.

| Method | Image Size | #Param. | ImageNet Top-1 Acc. |
|---|---|---|---|
| ViT-B/16 (Dosovitskiy et al., 2020) | $384^2$ | 86M | 77.9 |
| ViT-L/16 | $384^2$ | 307M | 76.5 |
| DeiT-S (Touvron et al., 2021) | $224^2$ | 22M | 79.8 |
| DeiT-B | $224^2$ | 86M | 81.8 |
| Vim-S (Zhu et al., 2024) | $224^2$ | 26M | 80.3 |
| Vim-B | $224^2$ | 98M | 81.9 |
| MANAR-256.32.8-S | $224^2$ | 28M | **80.7** |
| MANAR-256.32.8-B | $224^2$ | 108M | **82.3** |

## 4.2 KNOWLEDGE TRANSFER

The process of training a model from scratch might be prohibitive, especially as models reach hundreds of billions of parameters. When developing a new architecture, co-developing a method for transferring knowledge from existing trained models can lead to considerable reduction in training costs. Since transformer architecture serves as the backbone model for many applications, transferring knowledge from it is of great importance as it could drastically lower the barrier of new model's adoption. We developed a straightforward method to transfer knowledge from existing trained transformer architectures. Since our architecture (as depicted in Fig. 2) feature $q, k, v$ and $out\_proj$ weight matrices, which have a similar semantic meaning as the weights found in MHA, transferring knowledge from a trained transformer architecture requires copying those weight matrices. When training the knowledge-transferred MANAR model, we fix the copied weights (rendering them untrainable) and train only the remaining new weights. To illustrate the effectiveness of this method we tested it on two different applications: Image classification and Automatic speech recognition.

**Image classification** we loaded an existing DeIT model achieving 83.4% top-1 accuracy. We substitute every MHA layer with MANAR that has the same $q, k, v, out\_proj$ weight matrices. Each MANAR layer contains 256 memory cells, $ACR$ size of 32 and a context window length of 96. All the copied weight matrices (from the original DeIT) are fixed and do not train. We train the remaining weights for 20 epochs with exactly the same training setting as presented in Sec. 4.1. This setting achieves 83.1% top-1 accuracy. Both the reduction in the number of epochs, coupled with the fact that most of the model is no longer trainable, demonstrates a training time reduction of more than 99% compared to training MANAR from scratch.

Table 2: Word Error Rate (WER) on LibriSpeech standard dev/test sets. All models have the same 12-layer transformer encoder structure. The model output is decoded with the official 4-gram language model (Heafield, 2011)

| Model | dev-clean | dev-other | test-clean | test-other |
|---|---|---|---|---|
| wav2vec2.0 (Baevski et al., 2020) | 2.7 | 7.9 | 3.4 | 8.0 |
| HuBERT (Hsu et al., 2021) | 2.7 | 7.8 | 3.4 | 8.1 |
| data2vec (Baevski et al., 2022) | **2.2** | **6.4** | **2.8** | **6.8** |
| MANAR-256.64.8 | 2.3 | 6.7 | 2.9 | **6.8** |

**Automatic Speech Recognition** We use data2vec-base (Baevski et al., 2022) as a baseline model to transfer knowledge from, towards training a MANAR-256.64.8 model. We employ a local context window length of 128, capturing approximately 2.5 seconds of audio. All copied weights are fixed for training and the remaining weights are trained on 100 hours from LibriSpeech (Panayotov et al., 2015) using the CTC loss (Graves et al., 2006). We measure the resulting Word Error Rate on the LibriSpeech standard dev/test sets, compare with state-of-the-art solutions, and report the results as in Tab. 2. MANAR achieves state-of-the-art WER, confirming the effectiveness of the knowledge transfer method.

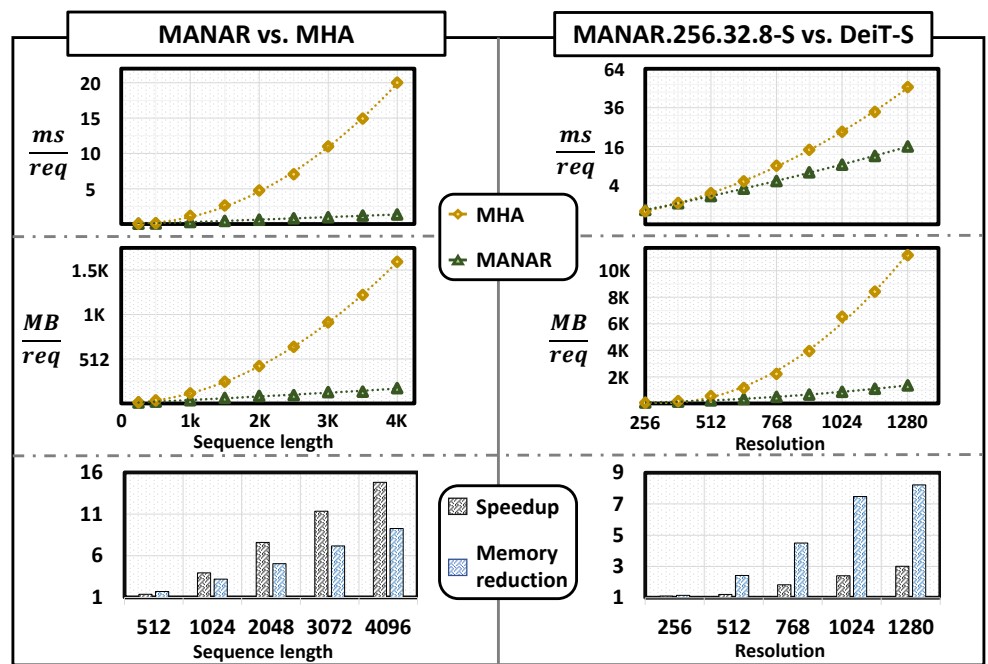

Figure 3: (left): Comparing the time and space complexity of MANAR against vanilla MHA; (right): Comparing the time and space complexity of MANAR-based DeiT-S (MANAR-256.32.8-S) against vanilla DeiT-S. Note the up right time measurements are made in quadratic scale to accommodate the quadratic effect of resolution on results.

### 4.3 PERFORMANCE COMPARISON

We profile MANAR against quadratic MHA under two conditions: single-layer latency and an end-to-end vision model. All measurements are performed on a single NVIDIA H100 GPU, with the batch size for each sequence length set to the maximum that could be accommodated in device HBM. Latencies report wall-clock averages over repeated runs, and memory refers to peak allocated GPU memory during inference. In the single-layer setting (left panels of Fig. 3), both layers use 12 heads of dimension 64, while MANAR employs 256 memory cells, an ACR of 32, and a local context window set to half the sequence length. At 256 tokens the two layers are essentially tied ($41.5\mu$s vs. $42.9\mu$s). At 2,048 tokens MANAR is already $7.6\times$ faster (0.62 ms vs. 4.74 ms) while cutting peak memory $5.0\times$, and at 4,096 tokens the gap widens to $14.8\times$ in latency (1.35 ms vs. 20.0 ms) and $9.3\times$ in memory. To verify robustness, we varied the ACR size between 16 and 512 and the memory size between 256 and 16K. Speedup and memory reduction figures remained essentially unchanged across the entire range, confirming that performance and memory gains are not sensitive to these hyperparameters.

In the end-to-end setting (right panels), replacing all MHA layers in DeiT-S with MANAR (MANAR-256.32.8-S) and setting local context window to half the sequence length yields slightly higher latency on $256\times256$ inputs (0.54 ms vs. 0.44 ms), but achieves a $2.0\times$ speed-up (7.15 ms vs. 14.6 ms) and $4.5\times$ memory reduction at $896\times896$, rising to $3.1\times$ faster (16.2 ms vs. 49.8 ms) and $8.2\times$ leaner at $1{,}280\times1{,}280$.

These findings highlight that although MANAR (with local context window which is half the sequence length) still scales quadratically in theory, the constant factor is significantly smaller than that of full MHA. If however the context window length were fixed to a constant, MANAR would achieve strictly linear scaling in both time and memory. In practice, even under the conservative $cw\_len = n/2$ setting, the reduction in constant factors consistently translates into large empirical gains in wall-clock time and memory footprint, enabling efficient execution at high sequence lengths and input resolutions.

### 4.4 MEASURING "*out-of-the-box thinking*"

To assess MANAR's ability to render representations that go beyond what can be obtained from a convex combination of input token values, we adopt the Convex Hull Membership (CHM) criterion: for a contextualized output vector $y_i \in \mathbb{R}^d$ and a set of input value vectors $\{v_1, \ldots, v_n\}$, CHM asks whether there exist non-negative coefficients

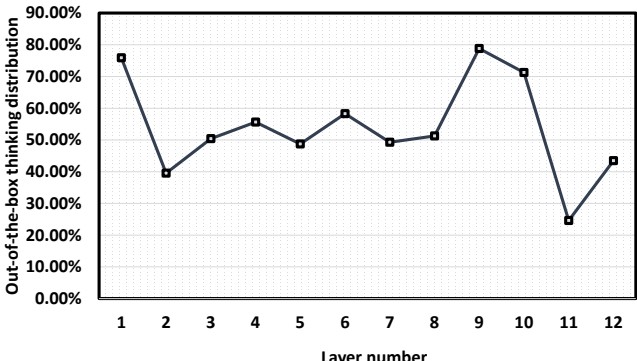

Figure 4: For each layer we sampled 10,000 token outputs and tested whether they lie outside the convex hull of their attended input values (Convex-Hull Membership criterion). The plot reports the resulting fraction per layer, revealing that more than half of the representations in most layers—and up to 80% in layers 9–10—cannot be expressed as any convex combination of the input tokens, confirming MANAR's capacity to synthesize novel, non-convex contextual information.

$\alpha_1, \ldots, \alpha_n$ summing to 1 such that: $y_i = \sum_{j=1}^{n} \alpha_j v_j$. If no such coefficients exist, $y_i$ lies outside the convex hull of the inputs, signaling that the layer has synthesized information that cannot be expressed as a weighted average of the original token values, hence not obtainable by MHA.

Such "*out-of-the-box thinking*" measurements are conducted on the MANAR-256.32.8 image classification knowledge transferred model. For each layer, we randomly select 10,000 output tokens $y_i$ as appears in Eq. 9, drawn uniformly across all heads. For each sample we collect the corresponding set of all input values $\{v_1, \ldots, v_n\}$. We then solve the CHM problem using linear programming as suggested by (Dantzig, 2016) .Fig. 4 plots, for each encoder layer, the fraction of outputs that fall outside the hull. Most layers exceed 50%, confirming that MANAR regularly synthesizes non-convex representations and thus contextualizes tokens in ways vanilla MHA cannot. The layer-wise pattern also reveals that reliance on memorized concepts is high at the very beginning, indicating that tokens lean on the Abstract Conceptual Representation (ACR) to inject global context that their narrow local windows cannot yet provide. As representations mature in mid-stack layers, local evidence becomes richer and the model needs the ACR less, so the out-of-hull fraction stabilizes near 55%. Late layers then show a sharp rise to 70–80%, reflecting a second surge of memory usage when the network synthesizes higher-level, task-specific abstractions, indicating reliance on memory concepts which serve as internalized ready-made perceptions representing memorized abstract features. Finally, the fraction drops in the topmost layers.

## 5   CONCLUSION

We introduced MANAR, a memory-augmented contextualization layer that complements local token interactions with a compact *Abstract Conceptual Representation*, functioning as a memory-centric mental image. Embedded as a drop-in replacement for multi-head attention, the layer reduces the theoretical cost of contextualization from quadratic to linear, delivers up to 14.8× inference-time speedup and 9.3× peak memory saving at 4K tokens, and matches—or slightly exceeds—state-of-the-art accuracy on ImageNet-1K and LibriSpeech. Convex-hull membership analysis further shows that more than half of MANAR's outputs transcend the span of the input values, providing quantitative evidence that the memory pathway synthesizes genuinely novel, high-level representations rather than merely re-weighting visible tokens. Looking ahead, we regard MANAR as a first step toward *concept-driven* sequence modeling. A natural next milestone is to explore its utility on language-centric benchmarks such as information-retrieval and open-domain question-answering, where the ability to inject global semantic concepts may prove even more valuable than in vision or speech. At the architectural level, we are adapting the layer for causal, autoregressive decoding by reformulating both memory retrieval and ACR construction to operate with strictly left-to-right visibility, thereby enabling deployment in long-context language generators. Finally, while the present work focuses on *read-only* memory retrieval, the human analogy that inspired MANAR also highlights the importance of continually updating stored concepts; ongoing efforts therefore target differentiable write-back policies that can learn when to allocate, consolidate, or overwrite memories without disrupting stable retrieval.

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

## A  APPENDIX

We leveraged Large Language Models (LLMs) solely as writing assistants. Their role was confined to: (1) re-phrasing and clarifying sentences for improved readability; (2) catching minor grammatical or typographical errors.

### A.1  COMPLEXITY ANALYSIS

We analyze the computational complexity of the MANAR layer compared to standard multi-head attention (MHA). Consider an input sequence $X \in \mathbb{R}^{n \times D}$, where $n$ is the sequence length, $D = hd$ is the hidden dimension, and $h$ is the number of heads with per-head dimension $d$.

**Conceptualization.** Each token is projected into query, key, and value spaces using linear projections, plus an additional projection into the ACR key space. The cost per head is $O(ndD)$ and across all heads $O(nD^2)$. This is identical to standard MHA and scales linearly in sequence length.

**Search pattern building.** Constructing $m$ search patterns requires cross-attention between $m$ mixer vectors and the $n$ tokens. For each mixer, similarity scores against all $n$ tokens cost $O(nd)$, followed by a weighted sum of token values costing another $O(nd)$. The overall cost per head is $O(mnd)$, independent of memory size $M$. Overall, across all heads the cost is $O(mnD)$.

**Memory retrieval.** Naively, computing similarity between each search pattern and all $M$ memory keys requires $O(mdM)$. With top-$k$ selection, this yields $O(mdM+mkd)$, which is prohibitive for large $M$. MANAR uses product-key factorization: each key is represented implicitly as the Cartesian product of two half-keys of size $\sqrt{M}$. This reduces the lookup cost to $O(md\sqrt{M})$ for similarity computation, plus $O(mk^2)$ for combining candidate pairs. This is the main complexity reduction, yielding sublinear scaling in $M$ compared to naive lookup. The overall complexity cost when taking into account all heads is $O(mD\sqrt{M})$.

**ACR construction.** Each of the $m$ retrieved memory concepts is contextualized by $n$ tokens through a softmax over token keys and values. This requires $O(mnd)$ multiplications. Across all heads, the cost is $O(mnD)$.

**Token contextualization.** Each token attends to (i) the $m$-sized ACR and (ii) its local window of length $L$. The complexity across all heads is $O(nD(m+L))$. For fixed $L$ and $m$, this is linear in sequence length. For $L = O(n)$ the scaling becomes quadratic, but with a constant factor that is substantially smaller than full MHA.

**Overall complexity.** Summing all stages, the MANAR layer has complexity

$$O(nD^2) + O(mnD) + O(mD\sqrt{M}) + O(nD(m+L)) = O\big(nD^2 + mnD + nLD + mD\sqrt{M}\big).$$

For fixed $m, L$, and moderate memory size $M$, this simplifies to $O(nD^2)$, i.e., linear in sequence length. In contrast, standard MHA has complexity $O(n^2D)$. Thus MANAR achieves linear dependence on $n$, controlled by $m$ and $L$, and avoids quadratic growth.

**Summary.** MANAR reduces complexity from quadratic in $n$ (MHA) to linear in $n$ with sublinear memory lookup costs in $M$. The overall complexity is governed by (i) the number of memory lookups $m$, (ii) the retrieval granularity $k$, and (iii) the local context window length $L$. Finally, to fully exploit the structure of MANAR operations, we implemented a custom Triton kernel that fuses computations, yielding substantial wall-clock speedups in practice.

