# OpenReview forum: "MANAR: Memory-augmented Attention with Navigational Abstract conceptual Representation"
_ICLR.cc/2026/Conference — Submitted to ICLR 2026_

### Official Review · Reviewer_6pDP · 2025-10-30

**Soundness:** 2
**Presentation:** 2
**Contribution:** 2
**Rating:** 2
**Confidence:** 2

**Summary:**

The paper introduces MANAR, a drop-in replacement for MHA that combines local token attention with a retrieved, fixed-size Abstract Conceptual Representation (ACR) from a trainable external memory. The goal is to mitigate the quadratic time/space cost of standard attention. Reported results include up to 14.8× latency and 9.3× peak-memory reductions at 4k tokens, “linear” scaling for fixed windows, ImageNet-1K top-1 = 82.3% (DeiT-B capacity), and LibriSpeech WER 2.9/6.8.

Although MANAR sounds promissing, the empirical evidence is not convincing. Experiments were conducted only on two tasks, including ImageNet classification and speech recognition. Missing comparison with other baselines e.g., sparse transformers, Linformer, or other linear attention methods. Experiments in the language domain would substantially strengthen the claim.

**Strengths:**

- The idea is clearly present: a unification of retrieved global context (ACR) with local attention to avoid all-pairs attention.
- Efficiency: Substantial wall-clock and HBM savings in microbenchmarks and end-to-end DeiT-S at large resolutions, with improvements growing with sequence length.
- MANAR enables quick adoption and a large reduction in trainable parameters/steps while retaining accuracy on vision and speech.

**Weaknesses:**

- **Modest accuracy gains:** On ImageNet-1K, improvements over DeiT-B are small (82.3% vs. 81.8%). For ASR, the paper claims SOTA, but test-clean 2.9 trails data2vec (2.8) and test-other is tied at 6.8.
- **Related work gaps:** While Linformer/Performer and long-sequence families (Mamba/RetNet, KV-cache management) are cited, several key lines are missing or under-discussed: sparse attention baselines, Swin/local-window ViTs, Transformer-XL/Compressive Transformer, standard retrieval-augmented modeling for LMs, and Memory Transformer.
- **Lack of empirical experiments:** See questions below.

**Questions:**

- How is the accuracy vs latency curve across different values of $L$?  How does the number of retrieved memory concepts affect the performance?
- The speedups rely on a custom fused Triton kernel. There is no ablation quantifying how much of the gain comes from kernel engineering rather than from the MANAR design itself.
- The modified paper layout appears to create extra space; please adhere to the venue’s formatting rules.

---

### Official Review · Reviewer_dhQL · 2025-11-09

**Soundness:** 1
**Presentation:** 1
**Contribution:** 1
**Rating:** 0
**Confidence:** 1

**Summary:**

Paper not reviewed due to formatting issues.

**Strengths:**

-

**Weaknesses:**

-

**Questions:**

-

---

### Official Review · Reviewer_LXgz · 2025-11-09

**Soundness:** 1
**Presentation:** 1
**Contribution:** 1
**Rating:** 0
**Confidence:** 4

**Summary:**

N/A, see ethics comment & 'Weaknesses' section

**Strengths:**

N/A, see ethics comment & 'Weaknesses' section

**Weaknesses:**

As pointed out at the beginning of the reviewing phase, the margins of the paper unfortunately appear to have been significantly altered, which allows more space than the original template.

I have to therefore recommend desk-rejection / rejection due to misuse of format.

**Questions:**

N/A, see ethics comment & 'Weaknesses' section

**Details Of Ethics Concerns:**

As pointed out at the beginning of the reviewing phase, the margins of the paper unfortunately appear to have been significantly altered, which allows more space than the original template.
(See 'official comments' and corresponding exchange with AC and other reviewer.)

I have to therefore recommend desk-rejection.

---

### Meta-Review · Area_Chair_KVdL · 2025-12-28

**Summary:**

The paper introduces MANAR, a linear-complexity alternative to multi-head attention that utilizes a trainable external memory for retrieving abstract conceptual representations to address scaling bottlenecks. While the proposal aims to improve computational efficiency, the technical assessment noted that the empirical gains were modest on standard benchmarks like ImageNet and LibriSpeech, and that the evaluation lacked necessary comparisons to relevant sparse attention baselines. However, the overriding consensus for rejection stems from a significant violation of submission policies regarding formatting. Multiple reviewers and the Area Chair identified that the paper’s margins were substantially altered to circumvent page limits, effectively allowing the authors to include more content than the standard template permits. Consequently, I recommend rejection primarily due to non-compliance with formatting standards, alongside the noted technical limitations.

**Reviewer Concerns:**

Outstanding concerns:
- The paper’s margins were substantially altered to circumvent page limits, effectively allowing the authors to include more content than the standard template permits.

**Reviewer Scores:**

I believe most reviewers would maintain their score if they had been able to participate fully in the discussion.

---

### Decision · Program_Chairs · 2026-01-26

Reject